# Rethinking Weight Decay for Robust Fine-Tuning of Foundation Models

**Junjiao Tian**
Georgia Institute of Technology
`jtian73@gatech.edu`

**Chengyue Huang**
Georgia Institute of Technology
`chuang475@gatech.edu`

**Zsolt Kira**
Georgia Institute of Technology
`zkira@gatech.edu`

## Abstract

Modern optimizers such as AdamW, equipped with momentum and adaptive learning rate, are designed to escape local minima and explore the vast parameter space. This exploration is beneficial for finding good loss basins when training from scratch. It is not necessarily ideal when resuming from a powerful foundation model because it can lead to large deviations from the pre-trained initialization and, consequently, worse robustness and generalization. At the same time, strong regularization on all parameters can lead to under-fitting. We hypothesize that selectively regularizing the parameter space is the key to fitting and retraining the pre-trained knowledge. This paper proposes a new weight decay technique, Selective Projection Decay (SPD), that selectively imposes a strong penalty on certain layers while allowing others to change freely. Intuitively, SPD expands and contracts the parameter search space for layers with consistent and inconsistent loss reduction, respectively. Experimentally, when equipped with SPD, Adam consistently provides better in-distribution generalization and out-of-distribution robustness performance on multiple popular vision and language benchmarks. Code available at `https://github.com/GT-RIPL/Selective-Projection-Decay.git`.

## 1 Introduction

Modern optimizers, such as Adam [1], LARS [2], and LAMB [3] usually include momentum and adaptive learning rates. They help optimizers avoid local minima and accelerate learning [4, 5] to explore wider parameter spaces. However, we hypothesize that this behavior is not always beneficial for fine-tuning from a *well* pre-trained foundation model, especially when fine-tuning a few layers is already sufficient for fitting the target data [6, 7, 8, 9]. Several prior works have found that unnecessary exploration will lead to large deviation from the initialization and worse robustness [10, 11], and constraining the deviation can improve a model's generalization on in-distribution (ID) data and robustness to out-of-distribution (OOD) data [12, 13, 14][1]. For example, L2-SP [13] imposes a regularization term on the distance between the current and pre-trained models. More recently, TPGM [10] and FTP [11] propose to learn different hard constraints for each layer. These new works have demonstrated impressive results on benchmarks. However, they are either difficult to tune, specialized to specific settings, or require significant computation and storage overhead. *This motivates us to ask whether a simple few-liner solution exists for this fundamental problem.*

---

[1]In this paper, ID generalization and OOD robustness refer to test accuracy on the fine-tuning distribution and robust accuracy on other shifted distributions, respectively.

38th Conference on Neural Information Processing Systems (NeurIPS 2024).

We propose re-examining the existing methods and summarizing their findings to find this solution. Starting from the simplest: L2-SP [13]. Specifically, L2-SP adds an L2 regularization term to the original objective function. Formally,

$$\mathcal{L}(\theta) = \tilde{\mathcal{L}}(\theta) + \frac{\lambda}{2}\|\theta - \theta_0\|_2^2 \qquad (1)$$

where $\theta$ denotes the model parameters, $\theta_0$ the initialization, $\tilde{\mathcal{L}}(\theta)$ the original objective function, and $\lambda$ the hyper-parameter for regularization strength. When $\theta_0 = \mathbf{0}$, L2-SP reduces to an ordinary weight decay. This simple method should be effective enough to constrain the model, as our experiments show it can reduce the deviation between the fine-tuned and pre-trained models (Sec. 4.1). However, it is held back by an important design choice: the penalty is always applied to all model parameters. Our empirical results identify that a large $\lambda$ prevents every layer from deviating too much and leads to poor fitting, while a small $\lambda$ cannot provide enough regularization. This significantly limits the otherwise effective design (Sec. 4.1). So, *what is missing in this algorithm?*

Recent works in robust fine-tuning and parameter-efficient fine-tuning (PEFT) have shown that customizing constraints for each layer and *selectively* choosing layers for fine-tunning can improve robustness [10, 11, 6]. Inspired by these findings, we hypothesize that selectively imposing the regularization to different layers is the key. Therefore, we propose a simple *selective* version of L2-SP weight decay: selective projection decay (SPD). This new algorithm innovates in two aspects: a **selection condition** and a **regularization strength ratio**. The former determines when to apply regularization to a layer, and the latter determines the strength of regularization for intuitive hyper-parameter tuning. Specifically, we derive the selection condition from hyper-optimization [15, 16, 17] by treating the condition as an optimizable parameter (Sec. 3.3), and the regularization strength ratio by re-writing L2-SP as a projection operation (Sec. 3.4). Intuitively, when the condition is met, the algorithm imposes *large* regularizations on selected layers. This allows the algorithm to avoid unnecessary deviation and simultaneously fit into the fine-tuning data. We test SPD on large-scale computer vision, and NLP benchmarks with popular foundation models and test ID and OOD performance on various distribution and domain shifts. SPD achieves SOTA performance while being much simpler than other competing methods. Our contributions are:

- We propose a selective projection decay, a selective variant of the popular L2-SP/weight decay regularization methods, for robust fine-tuning of large foundation models. We show that selectivity is important to make regularization effective.

- We conduct a detailed study of ID and OOD performance on image classification and semantic segmentation with natural distribution and domain shifts. SPD improves ID and OOD performance on these benchmarks.

- We show that SPD consistently improves the performance of PEFT methods (e.g. LoRA [7] and adapters [9]) on 8 common sense reasoning language tasks with LLaMA-7B (-13B).

## 2 Related Works

**Robust Fine-Tuning with Distance Regularization.** Constraining the distance or deviation between the fine-tuned and pre-trained models has been studied in several prior works. L2-SP [13] explicitly adds an L2 norm penalty on the deviation and shows improved ID generalization for fine-tuning. MARS-SP [12] studies different forms of norms as the penalty. It shows that the Matrix Row Sum (MARS) norm can be a superior alternative to the L2 norm. These two methods impose "soft" penalties and can be less effective [18]. Instead, LCC [18] proposes constraining the deviation through direct projection on the parameters, which also enforces a hard constraint on the Lipschitz continuity of the fine-tuned model. However, LCC is hard to tune because the projection radius is not an intuitive hyper-parameter. Furthermore, using a single projection constraint for all layers is not an ideal strategy [10]. More recently, TPGM [10] proposes to automatically learn the constraints in LCC during fine-tuning, customizing a different projection radius for each layer through a bi-level optimization scheme. FTP [11] further improves the computation efficiency of TPGM by adopting hyper-optimization [15, 16, 17] in its computation. Nevertheless, FTP is still difficult to control because hyper-optimization requires a secondary optimizer with additional optimization hyper-parameters, and the learned regularization can be too strong with no intuitive way to adjust. In contrast, SPD is a much simpler and more intuitive method, which can be implemented with just a few lines of code. The superior controllability makes SPD potentially applicable to more applications.

**Parameter Efficient Fine-Tuning (PEFT).** PEFT methods such as adapters [9, 8] and LoRA [7] have been proposed to reduce training memory usage and computation complexity. Recent works have found that PEFT methods also provide good robustness because they modify fewer parameters and retain more knowledge of the pre-trained models [11]. Surgical fine-tuning [6] concludes that fine-tuning a selective few layers can improve ID generalization. These new works motivate us to re-evaluate L2-SP and weight decay, often uniformly applied to all layers. We identify that the inferior performance of the simple methods is because of this uniformity, which exhibits a strong trade-off between fitting and regularization. Other robust fine-tuning methods, such as LP-FT [19] and FLYP [20], focus on feature distortion. We will review them in the Appendix 8.1.

**Other Robust Fine-Tuning Methods.** WiSE-FT [14] discovers that linearly interpolating between the fine-tuned and pre-trained models after fine-tuning can improve out-of-distribution robustness. This demonstrates that a closer distance to the pre-trained model can improve robustness. However, it only applies to models with zero-shot capabilities. Another orthogonal line of research for robust fine-tuning focuses on feature distortion. LP-FT [19] shows that fine-tuning with a randomly initialized head layer distorts learned features. It proposes a simple two-stage method to train the head layer first and then fine-tune the entire model. FLYP [20] shows that fine-tuning a foundation model using the same objective as pre-training can better preserve the learned features. Our contribution is an optimization method to penalize the derivation between the fine-tuned and pre-trained models explicitly during fine-tuning, which is orthogonal to them.

## 3 Methods

In this section, we first provide an overview of the Selective Projection Decay (SPD) method and then describe the intuition behind SPD with a numerical example. Finally, we provide a concrete mathematical motivation for our method's algorithmic design.

### 3.1 Selective Projection Decay (SPD)

**Formulation.** SPD is a regularization technique that penalizes significant deviation from the pre-trained model. We motivate the formulation from an existing method: L2-SP [13] (Eq. 1). L2-SP adds a distance penalty on the deviation between the fine-tuned and pre-trained models. The penalty is applied to all model parameters at all times. A large $\lambda$ prevents every layer from deviating too much and empirically leads to poor fitting, while a small $\lambda$ cannot provide enough regularization. This significantly limits the otherwise effective design. We propose a *selective* version of this simple technique: selective projection decay (SPD). We will examine L2-SP and SPD in Alg. 1 and Alg. 2.

**Notations.** We follow the notations in prior works [1, 21]. Let $m_t, v_t$ denote the moving average of the gradient and squared gradient, $\beta_1, \beta_2$ their hyper-parameters, and $\alpha$ the learning rate.

Alg. 1 shows the Adam optimizer with the L2-SP regularization in Eq. 1. The effects of the regularization are highlighted in blue, also shown in Eq. 2. Intuitively, the regularization leads to an interpolation-like equation[2]. If the product $\lambda\alpha = 1$, then $\theta_t \leftarrow \theta_0$ and if $\lambda\alpha = 0$, then $\theta_t \leftarrow \tilde{\theta}_t$, where $\theta_0$ and $\tilde{\theta}_t$ denote the initialization and the updated model *without* regularization.

$$\theta_t \leftarrow \tilde{\theta}_t - \lambda\alpha(\tilde{\theta}_t - \theta_0) \tag{2}$$

Alg. 2 shows the proposed SPD. There are two changes compared to Alg. 1.

- a condition,

$$c_t = -g_t^\mathsf{T}(\theta_{t-1} - \theta_0). \tag{3}$$

- a new interpolation-like equation with a multiplier, $r_t$, replacing the learning rate $\alpha$,

$$\theta_t \leftarrow \tilde{\theta}_t - \lambda r_t(\tilde{\theta}_t - \theta_0). \tag{4}$$

---

[2]Mathematically, this is not the precise formulation of L2-SP as written in Eq. 1. See the Appendix 8.3 for further discussion.

Compared to L2-SP, SPD only imposes a penalty when the *condition* is met ($c_t < 0$), and the strength of the penalty is controlled by a hyper-parameter $\lambda$ and an analytical quantity *deviation ratio* $r_t$, which we will introduce later.

**Algorithm 1:** Adam with L2-Regularization

**Initialize** $m_0 \leftarrow 0$, $v_0 \leftarrow 0$, $t \leftarrow 0$
**While** $\theta_t$ not converged
 $t \leftarrow t + 1$
 $g_t \leftarrow \nabla_\theta \tilde{\mathcal{L}}(\theta_{t-1})$
 $m_t \leftarrow \beta_1 m_{t-1} + (1 - \beta_1)g_t$
 $v_t \leftarrow \beta_2 v_{t-1} + (1 - \beta_2)g_t^2$
 **Bias Correction**
  $\widehat{m_t} \leftarrow \frac{m_t}{1-\beta_1^t}$, $\widehat{v_t} \leftarrow \frac{v_t}{1-\beta_2^t}$
 **Update**
  $\tilde{\theta}_t \leftarrow \theta_{t-1} - \frac{\alpha \widehat{m_t}}{\sqrt{\widehat{v_t}} + \epsilon}$
  $\theta_t \leftarrow \tilde{\theta}_t - \lambda \alpha (\tilde{\theta}_t - \theta_0)$

**Algorithm 2:** Adam with Selective L2-Reg.

**Initialize** $m_0 \leftarrow 0$, $v_0 \leftarrow 0$, $t \leftarrow 0$, $c_0 \leftarrow 0$
**While** $\theta_t$ not converged
 $t \leftarrow t + 1$
 $g_t \leftarrow \nabla_\theta \tilde{\mathcal{L}}(\theta_{t-1})$
 $m_t \leftarrow \beta_1 m_{t-1} + (1 - \beta_1)g_t$
 $v_t \leftarrow \beta_2 v_{t-1} + (1 - \beta_2)g_t^2$
 **Bias Correction**
  $\widehat{m_t} \leftarrow \frac{m_t}{1-\beta_1^t}$, $\widehat{v_t} \leftarrow \frac{v_t}{1-\beta_2^t}$
 **Update**
  $\tilde{\theta}_t \leftarrow \theta_{t-1} - \frac{\alpha \widehat{m_t}}{\sqrt{\widehat{v_t}} + \epsilon}$
 $c_t = -g_t^\mathsf{T}(\theta_{t-1} - \theta_0)$
 **If** $c_t < 0$ :
  $\theta_t \leftarrow \tilde{\theta}_t - \lambda r_t (\tilde{\theta}_t - \theta_0)$

## 3.2 Intuition Behind SPD

**SPD prioritizes layers with consistent improvement.** SPD adds regularization on layers that meet the condition $c_t < 0$ to slow their growth. The condition is determined by the sign of the inner product between two vectors. One vector is the negative gradient direction $(-g_t)$, i.e., the descent direction, and the other is the current progress direction $(\theta_{t-1} - \theta_0)$. The inner product between them measures the *alignment* between the *vanilla*[3] update direction and the progress so far. When the inner product is positive, the current progress direction generally points to a low loss region, and following it will lead to consistent loss reduction. Conversely, if the inner product is negative, the current progress direction will likely lead to a higher loss region, indicating inconsistent improvement. In this case, SPD will impose a penalty to slow down updates for those layers. Recall that modern optimizers use momentum to escape local minima and explore wider regions. Without this penalty, the model will likely head towards the higher loss region to overcome it. SPD chooses to slow down these layers and prioritizes layers with more consistent loss reduction. We will motivate this strategy in a principled manner and validate it in our experiments.

## 3.3 Deriving $c_t$ from Hyper-Optimization

Previously, we explained the intuition behind SPD. Specifically, we interpreted the condition $c_t$ as a measure of alignment and a test of update consistency. Nevertheless, there is a more profound reason why the quantity $c_t$ is a natural choice for selective regularization. In this section, we motivate SPD from a more mathematical perspective.

**Hyper-Optimization Setup.** Hyper-optimization is a technique to optimize hyper-parameters inside an optimizer [15, 16, 17]. They treat the hyper-parameters as trainable parameters and optimize them using another gradient-based optimizer. Let's start from the vanilla Adam with L2-SP algorithm (Alg. 1) and treat the regularization strength hyper-parameter $\lambda$ as a trainable parameter. To update $\lambda$, we need to obtain its gradient by taking a derivative w.r.t. $\lambda$ after applying it.

$$\nabla \lambda := \frac{\partial \mathcal{L}(\theta_t)}{\partial \lambda} = \frac{\partial \mathcal{L}(\theta_t)}{\partial \theta_t}^\mathsf{T} \frac{\partial \theta_t}{\partial \lambda} = \alpha * -g_{t+1}^\mathsf{T}(\tilde{\theta}_t - \theta_0). \tag{5}$$

**Selection Condition $c_t$.** Intuitively, if the quantity $\nabla \lambda$ is negative, applying the update in gradient descent will increase the value of $\lambda$, thus increasing the regularization strength of L2-SP. Conversely, a positive quantity will decrease the regularization strength. Therefore, the $\text{sign}(-g_{t+1}^\mathsf{T}(\tilde{\theta}_t - \theta_0))$ determines the change of regularization strength in the hyper-optimization of $\lambda$. Formally, we define the condition $c_t$ as,

$$c_t := -g_{t+1}^\mathsf{T}(\theta_t - \theta_0) \tag{6}$$

---

[3]the direction w/o momentum.

For memory efficiency, we use $(\theta_t - \theta_0)$ instead of $(\tilde{\theta}_t - \theta_0)$ because both vectors point in the same direction and won't affect the sign of $c_t$. This allows us to discard $\tilde{\theta}_t$. Otherwise, we need to keep an additional copy in memory. In summary, when $c_t < 0$, we apply a regularization for that layer as shown in Alg. 2. This calculation is done for each layer, and the regularization is selectively applied.

**Alternative Interpretation:** We just interpreted the selection condition $c_t$ in SPD as a measure of consistency between the current heading direction and the gradient direction. This perspective is more valid when the algorithm has accumulated some updates, i.e., $\|\theta_t - \theta_0\|_2 \gg 0$, and less justified when a heading has not been established at the beginning of training. To analyze this, we discuss the behavior SPD from the perspective of *stochastic* optimization when $\|\theta_t - \theta_0\|_2$ is small at the beginning of training in the Appendix 8.1.

### 3.4 Deriving $r_t$ from Projection

The selection condition $c_t$ determines when to apply regularization to which layers. However, one remaining question is the strength of regularization, which is not intuitive to tune. To overcome this, we introduced an analytical quantity, the deviation ratio $r_t$, in Eq. 2 and Alg. 1. In this section, we will motivate it from the perspective of projection.

**L2-SP is projection.** Projection onto a norm ball is common in constrained optimization. While L2-SP is not a constrained optimization problem, its operation bears similarity to projection. Suppose we project a model $\tilde{\theta}_t$ to an $\mathcal{L}_2$-norm ball with radius $\gamma$ centered around its initialization $\theta_0$. The equation of projection is the following,

$$\theta_p = \theta_0 + \frac{\gamma}{\max\{\gamma, \|\tilde{\theta}_t - \theta_0\|_2\}} * (\tilde{\theta}_t - \theta_0). \tag{7}$$

Equivalently, we can rewrite the equation as,

$$\theta_p = \tilde{\theta}_t - \left(1 - \frac{\gamma}{\max\{\gamma, \|\tilde{\theta}_t - \theta_0\|_2\}}\right) * (\tilde{\theta}_t - \theta_0). \tag{8}$$

Now, we can equate this equation to the highlighted L2-SP equation in Eq. 2 and Alg. 1, we can see that if $\lambda\alpha = \left(1 - \frac{\gamma}{\max\{\gamma, \|\tilde{\theta}_t - \theta_0\|_2\}}\right)$, the regularization is equivalent to projection with radius $\gamma$.

**Deviation Ratio $r_t$.** This equivalence inspires us to define a deviation ratio $r_t$:

$$r_t = \frac{\max\{0, \gamma_t - \gamma_{t-1}\}}{\gamma_t} \tag{9}$$

where $\gamma_t := \|\tilde{\theta}_t - \theta_0\|_2$ and $\gamma_t := \|\theta_{t-1} - \theta_0\|_2$ denote the current deviation (before regularization) and the previous deviation from the initialization $\theta_0$, respectively. We use $r_t$ in SPD (Alg. 2) to replace the learning rate $\alpha$ in L2-SP (Alg. 1) to make hyper-parameter ($\lambda$) tuning more intuitive. Specifically, suppose the hyper-parameter $\lambda = 1$, then the regularization in SPD is:

$$\theta_t \leftarrow \tilde{\theta}_t - \frac{\max\{0, \gamma_t - \gamma_{t-1}\}}{\gamma_t}(\tilde{\theta}_t - \theta_0) = \theta_0 + \frac{\gamma_{t-1}}{\max\{\gamma_{t-1}, \gamma_t\}} * (\tilde{\theta}_t - \theta_0). \tag{10}$$

Intuitively, with $\lambda = 1$, the regularization in SPD is equivalent to projection with a radius equal to the previous deviation if the current deviation is larger. In summary:

- **No regularization** ($\lambda = 0$): the projection radius is $\|\tilde{\theta}_t - \theta_0\|_2$, meaning no projection.

- **Weak regularization** ($1 \geq \lambda > 0$): the projection radius lies between $\|\tilde{\theta}_t - \theta_0\|_2$ and $\|\theta_{t-1} - \theta_0\|_2$. Within this range, all layers will expand or remain unchanged.

- **Strong regularization** ($\lambda > 1$): the projection radius lies between 0 and $\|\theta_{t-1} - \theta_0\|_2$. In this range, it's possible that regularized layers can contract.

We recommend starting with $\lambda = 1$ and adjusting the strength according to the specific needs.

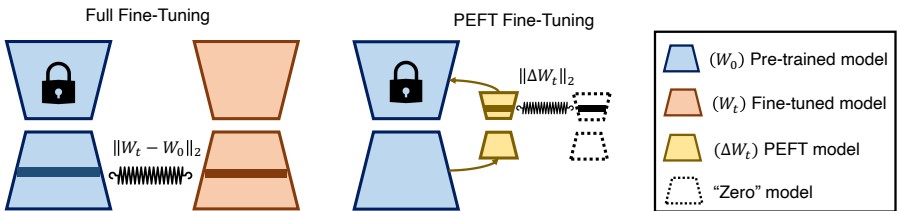

Figure 1: **Selective Projection Decay (SPD)** imposes regularization on layers selectively during fine-tuning. It regularizes $\|W_t - W_0\|_2$ for full fine-tuning and $\|\Delta W_t\|_2$ for PEFT fine-tuning.

### 3.5   Compatibility with PEFT methods.

As shown in Alg. 2, SPD retains a copy of the pre-trained model in memory. This adds additional memory requirements to the overhead of vanilla optimizers. While this is practical for moderate-sized models, as fine-tuning focuses more and more on large models, additional memory requirements become undesirable. Fortunately, in extremely large models, the prevalent fine-tuning strategy is parameter-efficient fine-tuning (PEFT), such as LoRA [7], series adapters [9], and parallel adapters [8]. SPD is naturally compatible with these methods without the additional memory. Intuitively, SPD selectively projects the current model towards the pre-trained initialization. PEFT methods generally initialize new parameters to add to the original model weights. To recover the behavior of SPD, we can instead project the new parameters towards the *origin*, equivalent to a selective version of regular weight decay, i.e., replacing $\theta_0$ with $\mathbf{0}$ in Alg. 2. Consequently, this does not require a memory copy of the pre-trained model. It consistently improves PEFT fine-tuning for large language models on common sense reasoning benchmarks in Sec. 4.4.

For example, LoRA decomposes a linear layer $h = W_t x$ into two components, where $h \in \mathbb{R}^{m \times 1}$, $W_t \in \mathbb{R}^{m \times n}$ and $x \in \mathbb{R}^{n \times 1}$ are the output, weights, and input of this layer.

$$h = W_t x = (W_0 + \Delta W_t)x \approx W_0 x + W_{up}W_{down}x \tag{11}$$

where $W_0 \in \mathbb{R}^{m \times n}$, $W_{up} \in \mathbb{R}^{m \times r}$ and $W_{down} \in \mathbb{R}^{r \times n}$ are the pre-trained model, up-projection and down-projection matrices. If $r \ll \min\{m, n\}$, $(W_{up}W_{down})$ is a low-rank approximation of $\Delta W_t$. To regularize the overall deviation $\|W_t - W_0\|_2$, it suffices to regularize $\|W_{up}W_{down}\|_2$ to be close to zero. In this case, SPD acts as selective weight decay on $W_{up}$ and $W_{down}$ individually.

In summary, we propose selective projection decay (SPD) to impose strong regularization on layers during fine-tuning selectively. As shown in Fig. 1, SPD regularizes the deviation of the fine-tuned model from the pre-trained model $\|W_t - W_0\|_2$ for full fine-tuning and the deviation from the origin $\|\Delta W_t\|_2$ for PEFT fine-tuning.

## 4   Experiments

We test Selective Projection Decay on a diverse set of benchmarks, architectures, and tasks to demonstrate its effectiveness. We will test both ID generalization and OOD robustness across various domain and distribution shifts.

**Image Classification.** We first analyze the behavior of SPD on conventional image classification datasets DomainNet [22] and ImageNet [23]. We use a CLIP ViT-Base model for both experiments as the pre-trained initialization [24]. Specifically, DomainNet consists of images from several domains with 345 classes. We fine-tune on one domain and test on all domains. ImageNet is a large-scale dataset with 1000 classes. We fine-tune on ImageNet and test on ImageNet and four variants, namely ImageNet-V2 [25], ImageNet-A [26], ImageNet-R [27], and ImageNet-S [28].

**Semantic Segmentation.** We further test SPD on the PASCAL-Context semantic segmentation dataset [29]. Following prior works [30, 11], we use a Swin ViT-Tiny [31], pre-trained on ImageNet-22K, and Segformer [32] segmentation architecture. To construct the OOD datasets, we follow the popular natural robustness literature [33] and apply four representative image corruptions (fog, defocus blur, Gaussian noise, and brightness) with 5 severity each. We fine-tune on the clean segmentation data and test on clean and corrupted data.

**Common Sense Reasoning.** Moreover, we show that SPD can benefit PEFT fine-tuning on large language models (LLMs). We use the Commonsense-170K dataset [34], which consists of training data from eight common sense reasoning benchmarks. Following the prior work [34], we fine-tune LLaMa-7B (-13B) [35] using LoRA [7], series adapters [9], and parallel adapters [8].

**Visual Question Answering.** Finally, we demonstrate SPD's superiority on multi-modal task. We use Google's recently released PaliGemma [36] pretrained on a broad mixture of large-scale vision-language tasks. We fine-tune on VQAv2 [37] and test on nine OOD datasets using LoRA [7]. For the near OODs, we evaluate on VQAv2's six variants, namely IV-VQA [38], CV-VQA [38], VQA-Rephrasings [39], VQA-CP v2 [40], VQA-CE [41] and AdVQA [42], which cover uni-modal, multi-modal and adversarial distribution shifts from VQAv2. We also include TextVQA [43], VizWiz [44] and OK-VQAv2 [45], which are constructed from different sources than VQAv2, as the far OOD datasets.

## 4.1 DomainNet Experiments

Table 1: Comparisons between AdamW and Adam-SPD on DomainNet. A pre-trained CLIP ViT-Base model is fine-tuned on each of the five domains in DomainNet and tested on all domains. Each row represents the evaluation of a model fine-tuned on a domain. ID performance is highlighted in blue. The last column shows the deviation of the final model from its initialization. Adam-SPD shows much better OOD performance with significantly less *Deviation* ($\|\theta_t - \theta_0\|_2$) than vanilla AdamW.

| Optimizer | Fine-tune Domain | Test Domains | | | | | | Statistics | | |
|---|---|---|---|---|---|---|---|---|---|---|
| | | Real | Clipart | Painting | Sketch | Quickdraw | Infograph | ID ↑ | OOD Avg. ↑ | Deviation↓ |
| AdamW | Real | 84.83 | 57.55 | 53.13 | 44.11 | 8.44 | 33.15 | 84.83 | 39.28 | 1.53 |
| L2-SP | | 82.33 | 53.35 | 51.82 | 42.04 | 8.21 | 30.84 | 82.33 | 37.25 | 0.70 |
| Adam-SPD | | **87.10** | **63.45** | **60.34** | **54.12** | **11.73** | **39.99** | **87.10** | **45.93** | **0.51** |
| AdamW | Clipart | 54.50 | 79.88 | 40.97 | 46.87 | 13.14 | 26.31 | 79.88 | 36.36 | 0.83 |
| L2-SP | | 55.73 | 79.67 | 41.61 | 47.12 | 11.51 | 26.51 | 79.67 | 36.50 | 0.70 |
| Adam-SPD | | **61.44** | **81.43** | **48.31** | **52.06** | **13.73** | **31.62** | **81.43** | **41.43** | **0.40** |
| AdamW | Painting | 55.62 | 46.64 | 74.90 | 40.56 | **8.55** | 26.18 | 74.90 | 35.51 | 0.81 |
| L2-SP | | 54.73 | 45.15 | 73.45 | 38.75 | 4.3 | 24.87 | 73.45 | 33.56 | 0.67 |
| Adam-SPD | | **60.66** | **52.43** | **77.77** | **47.81** | 6.38 | **30.84** | **77.77** | **36.92** | **0.38** |
| AdamW | Sketch | 45.02 | 52.97 | 39.70 | 72.26 | 15.16 | 18.79 | 72.26 | 34.33 | 0.95 |
| L2-SP | | 47.45 | 52.7 | 40.74 | 71.05 | 14.96 | 23.36 | 71.05 | 35.84 | 0.67 |
| Adam-SPD | | **52.81** | **57.39** | **46.90** | **74.00** | **15.77** | **24.35** | **74.00** | **39.44** | **0.40** |
| AdamW | Quickdraw | 3.08 | 10.12 | 1.66 | 9.61 | **68.68** | 1.04 | **68.68** | 5.10 | 1.72 |
| L2-SP | | 4.03 | 11.06 | 2.11 | 9.13 | 62.21 | 1.61 | 62.21 | 5.59 | 0.77 |
| Adam-SPD | | **18.72** | **24.36** | **12.77** | **20.61** | 66.81 | **7.06** | 66.81 | **16.70** | **0.58** |
| AdamW | Infograph | 51.49 | 42.46 | 37.20 | 35.46 | 6.02 | 52.71 | 52.71 | 34.53 | 0.85 |
| L2-SP | | 51.46 | 41.99 | 38.39 | 35.75 | 6.8 | 53.33 | 53.33 | 34.88 | 0.70 |
| Adam-SPD | | **58.29** | **48.25** | **46.00** | **43.38** | **7.88** | **56.36** | 56.36 | **40.76** | **0.36** |

In this section, we utilize the DomainNet benchmark to test our claims. Specifically, we will show that Adam-SPD consistently outperforms AdamW in OOD robustness across multiple domains, and this is due to a much smaller deviation from the pre-trained model. Furthermore, by sweeping across a range of hyper-parameters, we show that uniform regularization, such as L2-SP fails to provide adequate constraints, while Adam-SPD shows robust performance.

**Small deviation correlates with better OOD performance.** Earlier, we hypothesized that a significant deviation can lead to worse OOD performance. Theoretically, prior works [12, 11] have shown that large deviations from the initialization result in a large Liptschtz constant and, hence, worse robustness. In Tab. 1, we present a comprehensive study by fine-tuning a pre-trained CLIP model on different domains from DomainNet separately and reporting test results on all domains. Across all domains, Adam-SPD consistently achieves higher OOD performance and shows noticeably less deviation. This empirical result corroborates with prior works' findings and our hypothesis.

**Selective regularization exhibits stronger deviation-robustness correlation.** In Tab. 2, we compare the behavior of L2-SP and SPD using DomainNet. Specifically, we fine-tune a CLIP model on the Clipart domain (ID domain) and report performance on Clipart and other domains (OOD domains). In Tab. 2a, we observe that while L2-SP successfully restrains the model's deviation from its initialization, it does not effectively improve OOD performance. With a very large regularization, the ID performance deteriorates as well. On the contrary, SPD effectively restrains the model's deviation

Table 2: Comparisons between L2-SP and Adam-SPD. ID dataset: {clipart}, OOD datasets: {real, sketch, quickdraw, painting}. Selective regularization can effectively restrain model's deviation ($\|W_t - W_0\|_2$) and improve OOD robustness without significantly impacting ID robustness.

| Hyper-Parameter $\lambda$ | 1e-1 | 1e-2 | 6e-3 | 3e-3 | 1e-3 | 6e-4 | 3e-4 | 1e-4 | 1e-5 | 1e-6 | 1e-7 | 0.0 |
|---|---|---|---|---|---|---|---|---|---|---|---|---|
| Deviation | 0.03 | 0.14 | 0.18 | 0.24 | 0.34 | 0.39 | 0.46 | 0.53 | 0.58 | 0.58 | 0.58 | 0.59 |
| OOD | 14.90 | 37.20 | 39.43 | 40.52 | 41.13 | 41.76 | 40.52 | 41.26 | 41.35 | 41.73 | 40.62 | 41.34 |
| ID | 27.25 | 69.74 | 73.76 | 76.62 | 78.90 | 79.30 | 79.30 | 79.84 | 79.80 | 79.95 | 79.80 | 79.91 |

(a) L2-SP hyper-parameter ($\lambda$) sweep. Stronger regularizations (larger values) decrease deviation; however, they do not improve OOD performance and even deteriorate ID performance.

| Hyper-Parameter $\lambda$ | 2.1 | 1.9 | 1.7 | 1.5 | 1.3 | 1.1 | 0.9 | 0.7 | 0.5 | 0.3 | 0.1 | 0.0 |
|---|---|---|---|---|---|---|---|---|---|---|---|---|
| Deviation | 0.31 | 0.32 | 0.33 | 0.34 | 0.36 | 0.36 | 0.42 | 0.44 | 0.48 | 0.51 | 0.54 | 0.59 |
| OOD | 45.67 | 45.77 | 45.23 | 45.27 | 44.81 | 43.99 | 44.18 | 42.73 | 41.84 | 42.43 | 41.20 | 41.34 |
| ID | 81.21 | 80.76 | 81.25 | 80.67 | 81.11 | 79.89 | 79.57 | 80.00 | 79.92 | 80.26 | 80.00 | 79.91 |

(b) Adam-SPD hyper-parameter ($\lambda$) sweep. Stronger regularizations (larger values) decrease deviation, simultaneously improving OOD performance. The ID performance is not impacted significantly.

and significantly improves OOD performance while matching the best ID performance. Under SPD, the correlation coefficient between OOD performance and deviation is $-0.96$, which indicates a strong negative correlation between the two quantities, i.e., smaller deviation and higher OOD accuracy. This experiment shows that selective regularization is superior to uniform regularization.

**Training Details.** We use the vision transformer public repository for DEIT [46] to fine-tune all methods. We use $\lambda = 1$ for all Adam-SPD results in Tab. 1. More details are in Appendix 8.4.

## 4.2 ImageNet Experiments

Table 3: ImageNet Fine-Tuning Result using CLIP ViT-Base. SPD outperforms more complicated algorithms and beats L2-SP by $8.8\%$ by selectively imposing regularization.

| | ID | OOD | | | | Statistics | |
|---|---|---|---|---|---|---|---|
| | Im | Im-V2 | Im-Adversarial | Im-Rendition | Im-Sketch | OOD Avg. | Avg. |
| Zero-Shot | 67.68 | 61.41 | 30.60 | 56.77 | 45.53 | 48.58 | 52.40 |
| Vanilla FT | 83.66 | 73.82 | 21.40 | 43.06 | 45.52 | 46.98 | 54.29 |
| Linear Prob. | 78.25 | 67.68 | **26.54** | 52.57 | 48.26 | 48.76 | 54.66 |
| LP-FT | 82.99 | 72.96 | 21.08 | 44.65 | 47.56 | 46.56 | 53.85 |
| L2-SP | 83.44 | 73.2 | 20.55 | 43.89 | 46.60 | 46.06 | 53.54 |
| FTP | 84.19 | 74.64 | 26.50 | 47.23 | 50.23 | 49.65 | 56.56 |
| SAM | 83.67 | 73.66 | 20.48 | 42.98 | 45.70 | 45.71 | 53.30 |
| Adam-SPD | **84.21** | **74.83** | 25.42 | **49.09** | **51.18** | **50.13** | **56.95** |
| WISE-FT | 80.94 | 72.47 | 33.18 | 63.33 | 54.20 | 55.58 | 60.82 |
| WISE-SPD | **81.70** | **73.29** | **34.37** | **63.69** | **54.55** | **56.48** | **61.52** |

**SPD outperforms more complicated works on image classification.** Following the training recipe from the prior work [11], we fine-tune a CLIP ViT-Base model on ImageNet using Adam-SPD. We use the same hyper-parameters as the prior work and only adjust the regularization hyper-parameter in SPD. In Tab. 3, we observe that Adam-SPD provides the best ID performance (strong ID generalization) and best average OOD performance (strong OOD robustness) on four ImageNet variants. SPD achieves a level of competitive performance with just a few lines of code. SPD's simplicity and strong performance show that selective regularization is a fundamental improvement for robust fine-tuning.

**Training Details.** For Adam-SPD, we fine-tune the model with a learning rate of $3e - 5$ and $\lambda = 1.4$. The regularization hyper-parameter is found through cross-validation, and the model with the best ID validation accuracy is taken. More details are in Appendix 8.4.

## 4.3 PASACAL Dense Semantic Segmentation

Table 4: Pascal Semantic Segmentation Results with SWIN-Tiny transformers (ImageNet21K pre-trained). Performance is measured by mIoU↑. SPD improves OOD robustness compared to vanilla fine-tuning without regularization and L2-SP by $36.5\%$ and $5.8\%$, respectively.

| | ID | OOD | | | | Statistics | | |
| | Clean | Fog | Defocus | Gaussian | Brightness | OOD Avg. | ID Δ (%) | OOD Δ (%) |
|---|---|---|---|---|---|---|---|---|
| Vanilla FT | 66.03 | 56.72 | 38.04 | 23.21 | 58.03 | 44.00 | 0.00 | 0.00 |
| Adapter [9] | 71.85 | 69.36 | 50.94 | 37.43 | 68.26 | 56.50 | 8.82 | 28.40 |
| BitFit [47] | 70.31 | 67.00 | 46.39 | 30.61 | 66.22 | 52.56 | 6.49 | 19.44 |
| L2-SP [13] | 73.47 | 69.87 | 49.20 | 39.10 | 68.61 | 56.70 | 11.27 | 28.85 |
| MARS-SP [12] | 66.24 | 56.97 | 37.29 | 21.82 | 58.27 | 43.59 | 0.32 | -0.94 |
| LLRD [48] | 72.09 | 68.13 | 46.18 | 37.28 | 66.30 | 54.47 | 9.18 | 23.79 |
| TPGM [10] | 72.56 | 69.51 | 50.88 | 38.62 | 68.82 | 56.96 | 9.89 | 29.44 |
| FTP [11] | 73.79 | 71.10 | 52.63 | 40.25 | 69.81 | 58.45 | 11.76 | 32.83 |
| Adam-SPD | **74.27** | **71.74** | **53.41** | **44.17** | **70.92** | **60.06** | **12.47** | **36.50** |

**SPD outperforms more complicated works on semantic segmentation.** The same trend is observed on semantic segmentation in Tab. 4. Again, SPD achieves the best ID generalization and OOD robustness across four different corruptions. This shows that proper regularization is not only important for achieving strong ID generalization (performance on the test set) but also for strong OOD robustness (performance on distribution shifted test sets) to domains shift (Tab. 3) and distribution shift such as natural corruptions (Tab. 4). The model fine-tuned with SPD is consistently more robust across different levels of corruption and severity.

**Training Details.** For Adam-SPD, we fine-tune the model with a learning rate of $1e-4$ and $\lambda = 2.2$. The regularization hyper-parameter is found through cross-validation, and the model with the best ID validation accuracy is taken. More details are in Appendix 8.4.

## 4.4 LLaMA PEFT Fine-Tuning Experiments

Table 5: Accuracy comparison of LLaMA-7B (-13B) with different adapters and optimizers on eight commonsense reasoning datasets. SPD consistently improves fine-tuning performance on multiple PEFT methods across all datasets. Note that AdamW employs uniform weight decay by default.

| PEFT | LLM | Optimizer | BoolQ | PIQA | SIQA | HellaSwag | WinoGrande | ARC-e | ARC-c | OBQA | Avg. |
|---|---|---|---|---|---|---|---|---|---|---|---|
| Series | LLaMA$_{7B}$ | AdamW | 63.0 | 79.2 | 76.3 | 67.9 | 75.7 | 74.5 | 57.1 | 72.4 | 70.8 |
| | | Adam-SPD (1.0) | **68.3** | **80.4** | **77.4** | **81.6** | **79.7** | **79.4** | **63.5** | **78.4** | **76.1** |
| Parallel | LLaMA$_{7B}$ | AdamW | 67.9 | 76.4 | **78.8** | 69.8 | 78.9 | 73.7 | 57.3 | 75.2 | 72.3 |
| | | Adam-SPD (1.0) | **68.8** | **80.9** | 78.3 | **82.0** | **80.8** | **80.0** | **63.1** | **78.0** | **76.5** |
| LoRA | LLaMA$_{7B}$ | AdamW | 68.9 | 80.7 | 77.4 | 78.1 | 78.8 | 77.8 | 61.3 | 74.8 | 74.7 |
| | | Adam-SPD (0.7) | **69.1** | **82.8** | **78.9** | **84.8** | **80.7** | **80.9** | **65.8** | **79.2** | **77.8** |
| LoRA | LLaMA$_{13B}$ | AdamW | 72.1 | 83.5 | 80.5 | 80.5 | **83.7** | 82.8 | 68.3 | 82.4 | 80.5 |
| | | Adam-SPD (1.2) | **72.9** | **85.6** | **80.7** | **92.0** | **83.7** | **85.6** | **71.6** | **85.6** | **82.2** |

**SPD is compatible and consistently improves PEFT methods.** Previous experiments have shown that SPD imposes effective regularization for full fine-tuning. Furthermore, SPD can also improve the performance of PEFT methods. We fine-tune LLaMa-7B (-13B) models on the Commonsense-170k dataset [34]. As shown in Tab. 5, SPD consistently improves regular fine-tuning with AdamW, which uses a uniform weight decay for all tested PEFT methods. This demonstrates that selective regularization benefits full fine-tuning and PEFT fine-tuning. Combined with its simplicity, SPD can potentially improve generalization and robustness for more tasks in deep learning.

**Training Details.** We follow the training code released by a prior work [34]. We report the best performance from the original paper and compare them with Adam-SPD. More details are in Appendix 8.4.

### 4.5 Visual Question Answering (VQA) Experiments

Table 6: Visual Question Answering Result using PaliGemma-3B. SPD outperforms baselines across ID, near OOD and far OOD datasets using LoRA. Note that L2-SP reduces to Vinilla FT with AdamW under LoRA.

| | ID | Near OOD | | | | | | Far OOD | | |
|---|---|---|---|---|---|---|---|---|---|---|
| | | Vision | | Question | Answer | Multimodal | Adversarial | | | |
| | VQAv2 | IV-VQA | CV-VQA | VQA-Rephrasings | VQA-CP v2 | VQA-CE | AdVQA | TextVQA | VizWiz | OK-VQA |
| Zero-Shot | 54.42 | 63.95 | 44.72 | 50.10 | 54.29 | 30.68 | 30.46 | 14.86 | 16.84 | 28.60 |
| Vanilla FT(LoRA) | 86.29 | 94.43 | **69.36** | 78.90 | 86.21 | 71.73 | 49.82 | 42.08 | 22.92 | 48.30 |
| Linear Prob. | 78.24 | 87.83 | 63.87 | 69.61 | 78.48 | 61.66 | 42.90 | 29.61 | 18.80 | 42.27 |
| LP-FT(LoRA) | 85.97 | 93.30 | 65.93 | 76.49 | 86.16 | 72.73 | 45.68 | 31.41 | 19.01 | 43.27 |
| WiSE-FT(LoRA) | 71.36 | 85.06 | 64.55 | 66.42 | 70.89 | 48.74 | 43.95 | 36.98 | 22.41 | 42.35 |
| Adam-SPD(LoRA) | **87.39** | **95.25** | 68.85 | **79.48** | **87.27** | **73.52** | **50.90** | **43.56** | **23.05** | **50.11** |

**SPD shows competitiveness across ID, near OOD, and far OOD datasets on multimodal tasks.** Apart from uni-modal tasks, SPD outperforms other baselines on multi-modal tasks. We fine-tune PaliGemma-3B model on VQAv2 [37] dataset with LoRA. In Tab. 6, SPD improves vanilla fine-tuning and other robust fine-tuning methods, achieving best ID and average OOD performance w.r.t. distribution shifts across single modalities such as vision, question, answer and combinations of multiple modalities. We also show the performance evaluation for both near and far OOD datasets. SPD is consistently more robust under different types and degrees of distribution shifts.

**Training Details.** For Adam-SPD, we fine-tune the model with a learning rate of $1e - 3$ and $\lambda = 0.5$. The regularization hyper-parameter is found through cross-validation, and the model with the best ID validation accuracy is taken. More details are in Appendix 8.4.

## 5 Limitations

SPD is a selective regularization technique explicitly designed for fine-tuning. While it can be theoretically used for pre-training, it will likely lead to poor performance because it will hinder the training of some layers. For fine-tuning, it works well because the pre-trained foundation model is *assumed* to be a good initialization, and only small changes in a selected few layers can lead to a good local minimum. Furthermore, the level of performance gain depends on how well the foundation models are exposed to the fine-tuning and OOD data distributions during pre-training. For example, in the DomainNet experiment (Tab. 1), fine-tuning a CLIP ViT model on any other domain does not have reasonably good OOD robustness on the Quickdraw domain. One can deduce that Quickdraw is not well represented in the pre-training data of CLIP ViT.

## 6 Conclusion

Fine-tuning differs from training from scratch because it starts from a good initialization. Therefore, effective regularization is critical to retaining the knowledge of the pre-trained foundation model while fitting a model to the target distribution. We identified that 1) regularization is necessary to keep the fine-tuned model close to its initialization and maintain robustness; 2) uniform regularization can hurt model fitting if regularization is too strong. In this paper, we proposed selective projection decay (SPD), a selective version of the popular weight decay/L2-SP regularization method. With an additional few lines of code, SPD can be integrated into existing optimizers and performs selective regularization. It demonstrates superior regularization performance on different tasks and modalities in our experiments.

# 7 Acknowledgement

This work was supported by ONR grant N00014-18-1-2829.

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

# 8 Appendix

## 8.1 Extended Related Works

**Other Robust Fine-Tuning Methods.** WiSE-FT [14] discovers that linearly interpolating between the fine-tuned and pre-trained models after fine-tuning can improve out-of-distribution robustness. This demonstrates that a closer distance to the pre-trained model can improve robustness. However, it only applies to models with zero-shot capabilities. Another orthogonal line of research for robust fine-tuning focuses on feature distortion. LP-FT [19] shows that fine-tuning with a randomly initialized head layer distorts learned features. It proposes a simple two-stage method to train the head layer first and then fine-tune the entire model. FLYP [20] shows that fine-tuning a foundation model using the same objective as pre-training can better preserve the learned features. Our contribution is an optimization method to penalize the derivation between the fine-tuned and pre-trained models explicitly during fine-tuning, which is orthogonal to them.

## 8.2 Interpreting $c_t$ as an Early Layer Selection Criterion

In previous sections, we interpreted the selection condition $c_t$ in SPD as a measure of consistency between the current heading direction and the gradient direction. This perspective is more valid when the algorithm has accumulated some updates, i.e., $\|\theta_t - \theta_0\|_2 \gg 0$, and less justified when a heading has not been established at the beginning of training. This section discusses SPD from the perspective of *stochastic* optimization when $\|\theta_t - \theta_0\|_2$ is small at the beginning of training.

**Inner product of gradients captures gradient variance.** Modern deep learning models are trained by stochastic optimization techniques, e.g., mini-batch SGD, leading to stochasticity due to sampling. We first show that the inner product of gradients captures the variance of a sampling process. We invoke a common assumption in the convergence analysis of stochastic gradient descent [1, 49, 21]. Assuming that the stochastic gradient $g_t$ is a stationary process $\mathcal{G}$ over a short period, with a small step size, successive gradients, e.g., $g_t, g_{t+1}$, can be seen as samples drawn from the same distribution $\mathcal{G}$. Given two successive draws $g_1$ and $g_2$, we can approximate the first and second moment of $\mathcal{G}$.

$$\mathbb{E}\left[\|g\|^2\right] \approx \frac{1}{2}(\|g_1\|^2 + \|g_2\|^2), \qquad \|\mathbb{E}\left[g\right]\|^2 \approx \|\frac{1}{2}(g_1 + g_2)\|^2. \tag{12}$$

Define the *variation of gradients* as $Var(g) := \mathbb{E}\left[\|g - \bar{g}\|^2\right]$ [50, 51], where $\bar{g} := \mathbb{E}[g]$, we can show that

$$g_1^\mathsf{T} g_2 = 2\left(\frac{1}{4}\|g_1\|^2 + \frac{1}{4}\|g_2\|^2 + \frac{1}{2}g_1^\mathsf{T} g_2\right) - \frac{1}{2}\left(\|g_1\|^2 + \|g_2\|^2\right) \tag{13}$$

$$\approx \|\bar{g}\|^2 - \left(\mathbb{E}\left[\|g\|^2\right] - \|\mathbb{E}\left[g\right]\|^2\right) = \|\bar{g}\|^2 - Var(g)$$

**Remarks.** Eq. 13 shows that the inner product of two consecutive stochastic gradients, under certain assumptions, can be seen as the estimator for the difference between the gradient norm and the variance of gradients. When the inner product is negative, this indicates that the variance outweighs the magnitude of the gradient.

**SPD prioritizes layers with higher expected gain.** At the beginning of training, the heading direction $(\theta_1 - \theta_0)$ is dominated by early gradients. For example, at $t = 2$ the direction of $(\theta_1 - \theta_0)$ is the same as $-g_1$ in Adam. The sign of $-g_2^\mathsf{T}(\theta_1 - \theta_0)$ is the same as the sign of $g_2^\mathsf{T} g_1$. This shows that the condition $c_t$ captures the difference between gradient norm and gradient variance. With this interpretation, we show that $c_t$ reflects *expected performance gain* in stochastic optimization. To see it, we can invoke the descent lemma for SGD. For an L-smooth function $f(W)$ [50], the descent lemma for SGD states that,

**Lemma 1.** $\underbrace{\mathbb{E}[f(\theta_{k+1})] - f(\theta_k)}_{\textit{Expected Performance Gain}} \leq \underbrace{-\eta_k(1 - \frac{\eta_k L}{2})}_{\leq 0}\|\bar{g}_k\|^2 + \underbrace{\frac{\eta_k^2 L}{2}}_{\geq 0} Var(g_k),$

where $\eta_k \leq \frac{2}{L}$ is the learning rate.

**Remarks.** The term on the left hand side $\mathbb{E}[f(\theta_{k+1})] - f(\theta_k)$ is the expected performance improvement for each step. Ideally, this should be a negative quantity. On the right-hand side, we observe

that improvement depends on two quantities $\|\bar{g}_k\|^2$ and $Var(g_k)$. To lower the upper bound, we want a *large* $\|\bar{g}_k\|^2$ and a *small* $Var(g_k)$. According to the decoupling Eq. 13, the inner product between successive gradients approximates this proportionality. Consequently, a negative $c_t$ likely indicates a higher upper bound on the expected gain, meaning a smaller improvement. Therefore, SPD will prioritize layers with potentially larger expected gains.

## 8.3 Approximation in L2-SP

| **Algorithm 3:** (Ours) Adam with L2-Regularization | **Algorithm 4:** (Original) Adam with L2-Regularization |
|---|---|
| **Initialize** $m_0 \leftarrow 0, v_0 \leftarrow 0, t \leftarrow 0$ | **Initialize** $m_0 \leftarrow 0, v_0 \leftarrow 0, t \leftarrow 0$ |
| **While** $\theta_t$ not converged | **While** $\theta_t$ not converged |
| $\quad t \leftarrow t + 1$ | $\quad t \leftarrow t + 1$ |
| $\quad g_t \leftarrow \nabla_\theta \tilde{\mathcal{L}}(\theta_{t-1})$ | $\quad g_t \leftarrow \nabla_\theta \tilde{\mathcal{L}}(\theta_{t-1})$ |
| $\quad m_t \leftarrow \beta_1 m_{t-1} + (1-\beta_1)g_t$ | $\quad m_t \leftarrow \beta_1 m_{t-1} + (1-\beta_1)g_t$ |
| $\quad v_t \leftarrow \beta_2 v_{t-1} + (1-\beta_2)g_t^2$ | $\quad v_t \leftarrow \beta_2 v_{t-1} + (1-\beta_2)g_t^2$ |
| $\quad$ **Bias Correction** | $\quad$ **Bias Correction** |
| $\qquad \widehat{m_t} \leftarrow \frac{m_t}{1-\beta_1^t}, \widehat{v}_t \leftarrow \frac{v_t}{1-\beta_2^t}$ | $\qquad \widehat{m_t} \leftarrow \frac{m_t}{1-\beta_1^t}, \widehat{v}_t \leftarrow \frac{v_t}{1-\beta_2^t}$ |
| $\quad$ **Update** | $\quad$ **Update** |
| $\qquad \tilde{\theta}_t \leftarrow \theta_{t-1} - \frac{\alpha\widehat{m_t}}{\sqrt{\widehat{v}_t}+\epsilon}$ | $\qquad \tilde{\theta}_t \leftarrow \theta_{t-1} - \frac{\alpha\widehat{m_t}}{\sqrt{\widehat{v}_t}+\epsilon}$ |
| $\qquad \theta_t \leftarrow \tilde{\theta}_t - \lambda\alpha(\tilde{\theta}_t - \theta_0)$ | $\qquad \theta_t \leftarrow \tilde{\theta}_t - \lambda\alpha(\theta_{t-1} - \theta_0)$ |

The Adam with L2-SP Regularization algorithm in the main paper is not the precise mathematical implementation of the original formulation written in Eq. 1. To see the difference, we compare ours and the accurate implementation here in Alg. 3 and Alg. 4. In our implementation, we replaced $\theta_{t-1} - \theta_0$ (Alg. 4) with $\tilde{\theta}_t - \theta_0$ (Alg. 3). This is done intentionally to improve memory efficiency when transitioning to the selective version (see Adam-SPD in Sec. 3.3). We can make the following substitution to see how this modification changes computation. Starting from our implementation,

$$\theta_t = \tilde{\theta}_t - \lambda\alpha(\tilde{\theta}_t - \theta_0) \tag{14}$$
$$= \tilde{\theta}_t - \lambda\alpha(\theta_{t-1} - \frac{\alpha\widehat{m_t}}{\sqrt{\widehat{v}_t}+\epsilon} - \theta_0)$$
$$= \tilde{\theta}_t - \lambda\alpha(\theta_{t-1} - \theta_0) + \lambda\alpha^2 \frac{\widehat{m_t}}{\sqrt{\widehat{v}_t}+\epsilon}.$$

We can further combine the $\lambda\alpha^2 \frac{\widehat{m_t}}{\sqrt{\widehat{v}_t}+\epsilon}$ into the update of $\tilde{\theta}_t$. The new $\tilde{\theta}_t$ is

$$\tilde{\theta}_t = \theta_{t-1} - \frac{\alpha\widehat{m_t}}{\sqrt{\widehat{v}_t}+\epsilon} + \lambda\alpha^2 \frac{\widehat{m_t}}{\sqrt{\widehat{v}_t}+\epsilon} \tag{15}$$
$$= \theta_{t-1} - (1-\lambda\alpha)\frac{\alpha\widehat{m_t}}{\sqrt{\widehat{v}_t}+\epsilon}$$

Therefore, our implementation of L2-SP adds a minor additional dampening of the learning rate $\alpha$ by a factor of $(1-\lambda\alpha)$.

*What if we followed the original implementation of L2-SP as in Alg. 4?* This would change the condition $c_t$ in the main paper (Eq. 3) to

$$c_t = -g_t^\mathsf{T}(\theta_{t-2} - \theta_0). \tag{16}$$

At the current time step $t$, we only have access to the parameters $\theta_{t-1}$ from the previous step $t-1$. To calculate the $c_t$ in Eq. 16, we would have to store the weights from two steps back in memory. This increases memory consumption of the algorithm. As we have shown in Eq. 15, our implementation only differs from the original implementation slightly but reduces memory consumption. Therefore, we decided to make this approximation.

## 8.4 Training Details

**DomainNet.** We use the vision transformer public repository for DEIT [46] to fine-tune all methods. Standard augmentations are used for all: weight-decay (0.1), drop-path (0.2) [52], label-smoothing (0.1) [53], Mixup (0.8) [54] and Cutmix (1.0) [55]. The learning rate is $2e-5$ and trained for 60 epochs for Tab. 1 and 30 epochs for Tab. 2. We use $\lambda = 1$ for all Adam-SPD results in Tab. 1. We use 1 A40 GPU for each experiment.

**ImageNet.** The same procedure as the DomainNet experiment is used for training the ImageNet models. Standard augmentations are used for all: weight-decay (0.1), drop-path (0.2) [52], label-smoothing (0.1) [53], Mixup (0.8) [54] and Cutmix (1.0) [55]. We fine-tune all methods for 30 epochs and use the best hyper-parameters reported by the prior work [11]. For Adam-SPD, we fine-tune the model with a learning rate of $3e-5$ and $\lambda = 1.4$. The regularization hyper-parameter is found through cross-validation, and the model with the best ID validation accuracy is taken. We use 2 A40 GPUs for each experiment.

**Pascal Segmentation.** We follow the training code released by a prior work [30]. We fine-tune all methods for 60 epochs and use the best hyper-parameters reported by the prior work. For Adam-SPD, we fine-tune the model with a learning rate of $1e-4$ and $\lambda = 2.2$. The regularization hyper-parameter is found through cross-validation, and the model with the best ID validation accuracy is taken. We use 4 2080Ti GPUs for each experiment.

**Commonsense-170K.** We follow the training code released by a prior work [34]. We report the best performance from the original paper and compare them with Adam-SPD. For Adam-SPD, we fine-tune the model with an identical hyper-parameter setup as the released code and only adjust the regularization strength $\lambda$. The regularization hyper-parameter is found through cross-validation, and the model with the best ID validation loss is taken. We use 1 A40 GPU for each experiment.

**Visual Question Answering.** We follow the LAVIS [56] public repository to fine-tune all methods. We use the PaliGemma [36] model pretrained with $224 * 224$ input images and $128$ token input/output text sequences and fine-tune with the precision of bfloat16. Standard hyper-parameters are used for all: learning rate ($1e-3$), weight-decay ($1e-4$), optimizer (AdamW), scheduler (Linear Warmup With Cosine Annealing), warm-up learning rate ($1e-4$), minimum learning rate ($1e-4$), accumulation steps (2), beam size (5). The model is trained for 10 epochs with a batch size of 16 for Tab. 6. For LoRA [7], we limit our study to only adapting the attention weights and freeze the MLP modules for parameter-efficiency, specifically apply LoRA to $W_q, W_k, W_v, W_o$ with $r = 8$ in Tab. 6. We use $\lambda = 0.5$ for SPD. The regularization hyper-parameter is found through cross-validation, and the model with the best ID validation accuracy is taken. We use 8 A40 GPU for each experiment.

