# OpenReview forum: "Rethinking Weight Decay for Robust Fine-Tuning of Foundation Models"
_NeurIPS.cc/2024/Conference — NeurIPS 2024 poster_

### Official Review · Reviewer_33UR · 2024-07-12

**Soundness:** 3
**Presentation:** 2
**Contribution:** 3
**Rating:** 5
**Confidence:** 4

**Summary:**

The paper proposes a method for robust fine-tuning of deep networks. The proposed method is a variant of L2-SP. The original formulation of L2-SP applied a uniform L2 penalty on the difference between the weights of the pre-trained model and the fine-tuned model. The proposed method (L2-SPD) aims to make this penalty non-uniform for different layers. This is achieved by only applying weight decay to layers where the direction of descent and weight decay are sufficiently similar. The efficacy of the method is demonstrated over various image classification, segmentation and reasoning tasks, showing better ID and OOD performance than baselines.

**Strengths:**

* The projection interpretation is very interesting.
* The exposition of the method and its insights is very nice.
* The insight behind the method is simple, but the method is widely applicable for an important problem.
* The method is well compatible with LoRA style approaches, increasing its practical applicability.
* The method is stable across a wide range of hyper-parameters.
* Wide range of benchmarking is performed, and the empirical results are strong.

**Weaknesses:**

There are some minor issues with the presentation -
* It would be nice to have a short preliminaries section to define all notations used in the paper, along with their dimensions.
* It is not clear from Alg 2 how SPD is selective to different layers. It appears that $\theta_t$ represents the entire model’s parameters, and $c_t$ is a single scalar for the entire model. Hence, it seems like the “weight decay” only happens for the entire model if the total gradient is in the opposite direction of the weight decay. I might be incorrect here, but I am unable to see where the group structure comes in (possibly $c_t$ is separate for each layer?)

Apart from this, some of the experiments have the following weaknesses -
* The Himmelbau example is not fully clear to me either. It would be interesting to compare the trajectories against L2-SP to actually understand the “selective penalization”, since regular L2-SP will also penalise deviations similarly. I tried running the code that the authors provided, and with some tuning of the hyper-parameters I was able to get similar trajectories with L2-SP as well.
* Lack of ablation studies on the two components of the method. It would be good to understand the gains from selective projection and adaptive regularization separately as well.
* I am unable to understand some of the trends in the results on ImageNet. It is not immediately clear to me how regularizing the model to minimize distance from the Zero-Shot model can outperform both the Zero-Shot model and the vanilla FT model on one domain (Im-Sketch), but is in between the two on another domain (Im-Rendition). I had the intuition that L2-SPD will pull the weights to be closer to the Zero-shot model, hence inheriting its robustness and biases, but I could be mistaken.
* Minor - The caption for Table 3 states that "SPD... beats L2-SP by 8.8%", however, I cannot understand how this was computed.
* Minor - While the paper compares against L2-SP, it misses Elastic Weight Consolidation, a similar method using Fisher information, which also adaptively sets the regularization parameters for different layers.

**Questions:**

* Is there a way to disentangle the gains from selective projection and adaptive regularization? Related to this, how often does the selective projection kick in in practice, and how does it change as training progresses (this is not crucial to the paper, but is a good-to-have analysis)?
* I am unable to understand some of the trends in the results on ImageNet. It is not immediately clear to me how regularizing the model to minimize distance from the Zero-Shot model can outperform both the Zero-Shot model and the vanilla FT model on one domain (Im-Sketch), but is in between the two on another domain (Im-Rendition). I had the intuition that L2-SPD will pull the weights to be closer to the Zero-shot model, hence inheriting its robustness and biases, but I could be mistaken.
* Can this technique of selective regularization be done at a more granular level beyond layers?
* In section 4.1, why are all domainnet domains not used, omitting the infograph domain? The paper will be helped by a justification of this.
* For table 2a, the range of deviation is quite different for L2-SP and L2-SPD. While L2-SPD does dominate performance for similar deviations, I wonder if it is possible to use even larger $\lambda$ and make the deviation very low? Will the same correlation with OOD accuracy still hold?
* Hyper-parameter selection is not clear for ImageNet. Was k-fold cross validation used? Is the ID val used to select hyper-parameters?
* On quickdraw, will more aggressive regularization yield better OOD accuracy? Since the pretrained model is supposed to be closer to other domains?
* For ImageNet, what is the linear layer for final classification?
* How does the method compare against a simpler baseline like Wise-FT?

**Limitations:**

The authors have addressed limitaions of their method, pointing out worse performance if the base model is not aligned with the OOD distribution.

---

> ### Author Response · Authors · 2024-08-06
>
> We appreciate the reviewer's taking the time to examine the paper and the provided code closely. We clarified your concerns and added Infograph experiments as requested. If anything is not clear, we are happy to discuss it during the rebuttal period.
>
> **The Himmelbau example.**
>
> We appreciate the reviewer's effort in running our code! In this simple example, you are correct that regular L2-SP would generate a similar trajectory. However, to see the difference between SPD's selective regularization and L2-SP, we suggest that the reviewer print out which direction is regularized during the optimization process.
>
> Add the following printing statement to line 40 in the code before the return.
> * print(self.condition_buffer)
> * return new_param_list
>
> The condition buffer contains the selection condition $c_t$ for the X and Y directions. Whenever an entry is negative, regularization is applied. This demonstrates the underlying mechanism for SPD.
>
> **Is there a way to disentangle the gains..?**
>
> Sorry for the confusion! SPD introduces only one component that affects performance, selective projection $c_t$ (Section 3.3),  which selectively applies regularization during fine-tuning. The selectivity is the only complement that provides the performance gain.
>
> Section 3.4 introduces a re-parametrization of the regular L2 regularization as a projection for better hyper-parameter tuning (see a comment from the reviewer Ex5N). It's a different way of writing L2 regularization and does not introduce performance gain.
>
> Selectivity kicks in very often. However, it is hard to quantify, and it changes depending on the task, model, data, optimization hyper-parameters, etc. An analysis of the selectivity's behavior will be insightful.
>
> **Trends in the results on ImageNet.**
>
> This is a very good question. Indeed, L2-SPD will pull the weights closer to the zero-shot model, as shown in Table 1 and Table 2 in the main paper, thus inheriting its robustness and biases.
>
> However, the extent of performance gain has a more subtle answer. It depends on the interplay between the pre-trained model, the fine-tuning (in-distribution) dataset, and the OOD datasets. Specifically, the ID dataset can provide useful information to the OOD datasets. In this benchmark, the useful information is that all datasets share the same label space. If fine-tuning leads to more gain, it's possible that after fine-tuning (assuming with proper regularization), the fine-tuned model can outperform both zero-shot and vanilla FT.
>
> **Can this technique of selective regularization be done at a more granular level beyond layers?**
>
> Yes, theoretically, the method can be applied at a parameter level. However, the storage cost is significantly increased because we would need to store a different $c_t$ for each parameter in the model.
>
> **In section 4.1, why are all domainnet domains not used, omitting the infograph domain?.**
>
> Infograph was omitted because none of the methods achieved good ID performance on this domain. We added the domain back to answer the reviewer's question (Table 1 in the rebuttal PDF). The superiority of SPD remains unchanged with the addition of this domain.
>
> **Larger regualrization and make the deviation very low? Will the same correlation with OOD accuracy still hold?**
>
> We have tried larger hyper-parameters. At extremely large values, the model will eventually underfit and perform poorly on ID and OOD data like L2-SP. We intend to show that L2-SPD outperforms L2-SP for similar deviations as the reviewer suggested.
>
> **Hyper-parameter selection is not clear for ImageNet.**
>
> We used the ID validation dataset following prior works. ImageNet-V2 can be seen as the clean test set for ImageNet.
>
> **On quickdraw, will more aggressive regularization yield better OOD accuracy?**
>
> Yes, we conducted a similar experiment as in Table 2 in the main paper. More aggressive regularization yields better OOD on quickdraw. We swept hyper-parameters from 1.0 to 2.3 and calculated the correlation coefficient between the hyper-parameters and OOD performance. The correlation is 0.88, which means a robust positive correlation between large regularization and good OOD performance.
>
> **Linear layer for For ImageNet**
>
> Because we used a CLIP ViT-Base model, we used a zero-shot linear head to initialize and fine-tune the linear layer. This practice follows from prior works such as WISE-FT.
>
> **Comparison with Wise-FT?**
>
> Technically, WISE-FT is an orthogonal method. Therefore, SPD should further improve its performance. To validate this, we added experiments combining WISE-FT and SPD in our Imagenet experiments (Table 1 in the rebuttal PDF). Our results show that SPD further improves the performance of Wise-FT. It's also worth pointing out that SPD is more general than WISE-FT, which only applies to models capable of zero-shot. It does not apply to the segmentation experiment (Table 4 in the main paper), whereas SPD does not have this constraint.

---

> > ### Comment · Reviewer_33UR · 2024-08-12
> > **Official Comment**
> >
> > I thank the authors for their response. The response has addressed most of my concerns.
> >
> > I believe that incorporating these into the paper will greatly improve the presentation, and hence I keep my rating of acceptance.

---

### Official Review · Reviewer_Jrpf · 2024-07-13

**Soundness:** 4
**Presentation:** 3
**Contribution:** 4
**Rating:** 6
**Confidence:** 4

**Summary:**

This paper proposes a new weight decay technique to adapt foundation models to target tasks, focusing on fitting the target data while maintaining the pre-trained knowledge. Specifically, the method, Selective Projection Decay (SPD), selectively imposes a strong penalty on certain layers while allowing others to change freely. Experimentally, the method consistently provides better in-distribution generalization and out-of-distribution robustness across multiple popular vision and language benchmarks.

**Strengths:**

- This is a well-motivated approach for fine-tuning foundation models on target tasks. By selectively imposing regularization on certain layers, it better fits the target data while retaining the pre-trained knowledge.
- The method has also proven to be effective for parameter-efficient fine-tuning, which is particularly beneficial.
- Extensive experiments conducted on both vision and language benchmarks demonstrate the method's strong performance.

**Weaknesses:**

There are no major weaknesses, please respond to my questions and issues.

**Questions:**

- Why is L2-SP not compared in the experiments shown in Tables 1 and 5? The authors are requested to provide the results for this method as well.
- Tables 2a and 2b should be converted into figures to enhance readability.
- Minor Typo: On line 8, "retraining" should be corrected to "retaining".
- Several of the results are wrongly highlighted in Table 3.
- The appendix sections referred in the paper are missing.

**Limitations:**

The limitation of this work is clearly highlighted: the method's performance depends on the quality of the foundation model.

---

> ### Author Response · Authors · 2024-08-06
>
> We thank the reviewer for the positive comments and pointing our our typos. We clarified your concerns and added L2-SP experiments as requested. If anything is unclear, we are happy to discuss it during the rebuttal period.
>
>
> **Why is L2-SP not compared in the experiments shown in Tables 1 and 5? The authors are requested to provide the results for this method as well.**
>
> * Table 1: Originally, we designed this table to present the improvement over a vanilla AdamW. For the rebuttal, we added L2-SP to Table 1 (Table 1 in the rebuttal PDF). SPD outperforms L2-SP in all domains.
>
> * Table 5: In the LLaMA PEFT experiments, we compared our method with Adam $+$ weight decay, as indicated in the table title. In this PEFT setting, L2-SP reduces to weight decay, as explained in section 3.5. In other words, the L2-SP baseline is already included.
>
>
> **Tables 2a and 2b should be converted into figures to enhance readability.**
>
> Thank you for the great suggestion. We will update the two tables to heat maps for better readability.

---

> > ### Comment · Reviewer_Jrpf · 2024-08-13
> >
> > Thank you to the authors for addressing my questions. I agree with the other reviewer that the theoretical perspective is lacking in this work. However, I believe the empirical results highlight the significance of the study, so I will keep my score.

---

### Official Review · Reviewer_CVio · 2024-07-13

**Soundness:** 3
**Presentation:** 3
**Contribution:** 2
**Rating:** 5
**Confidence:** 4

**Summary:**

The authors propose Selective Projection Decay, a modification on the AdamW optimizer where a regularization penalty is applied that controls for an overall drift from the prior weights such that as long as the overall shift is controlled for but individual layers can potentially adapt differently. The authors provide experiments on OOD generalization tasks and PEFT language modelling fine-tuning.

**Strengths:**

- The explanation of the method is clear and detailed. The method itself is straightforward and intuitive to understand.
- There are experiments on a number of different tasks and domains that show some improvement upon prior methods in terms of raw performance metrics.

**Weaknesses:**

- The way the authors frame it, there appears to be limited novelty with the proposed method, with it appearing to simply involve an additional conditional check before applying a potential regularization penalty that has been explored quite a bit in existing literature. I do think that there is something of merit here, however it's hard to pinpoint it under the current framing of the work.
- The statement "We recommend starting with  $\lambda= 1$ and adjusting the strength according to the specific needs" is rather unsatisfying and doesn't provide the reader with a good idea of whether or not $\lambda$ is easy to tune. This can probably be resolved by adding additional analysis along this end compare do just Table 2.
- Experimentally, the results feel somewhat lacking and somewhat misleading. For example, in the Domain-Net experiments, the only comparison is with AdamW, which inherently should lack the exploration capabilities the authors are attempting to account for. There are quite a few baselines that are missing as a result in my opinion, such as sharpness aware minimization (SAM) and/or other exploration based optimizers.
- For language modelling, it would be more useful to show performance over a multitude of models and sizes, as the results appear to suggest for example that the gap between AdamW and Adam-SPD narrows quite significantly between sizes 7B and 13B, as well as when the PEFT method changes.

**Questions:**

- I think a more interesting way to use the proposed method would be to have $\lambda$ to be a parameter or layer-specific hyper-parameter rather than a global hyper-parameter. This way, having individual $c_t$ for different components might make the method more robust and in fact help the model adapt relevant weights while keeping specific weights close to the pre-trained model. In many cases, different layers can begin to exhibit different functionalities and therefore there might be more use in having some layers change more than others based on the features that might be captured there.
- Although not a deal-breaker, I would appreciate a perhaps more in-depth analysis of the language modelling experiments, in particular without PEFT. While I understand the relationship with PEFT and the proposed optimizer, I believe that it would be quite valuable to show how AdamW-SPD works even under naive fine-tuning scenarios.

---

> ### Author Rebuttal · Authors · 2024-08-06
>
> We really appreciate the thoughtful comments, especially on suggesting a layer-specific hyper-parameter configuration. This was what we thought would be interesting as well. In the response, we provided clarifications to your concerns and a new experiment. If anything is not clear, we are happy to discuss it during the rebuttal period.
>
> **I think a more interesting way to use the proposed method would be to have to be a parameter or layer-specific hyper-parameter rather than a global hyper-parameter. This way, having individual for different components might make the method more robust and in fact help the model adapt relevant weights while keeping specific weights close to the pre-trained model. In many cases, different layers can begin to exhibit different functionalities and therefore there might be more use in having some layers change more than others based on the features that might be captured there.**
>
> Sorry for the confusion! SPD is a layer-specific method, as the reviewer suggested. SPD has a different selecting condition $c_t$ for each layer. In other words, algorithm 2 is displayed for each layer. SPD behaves precisely like the reviewer suggests, and we will update the writing to clarify this.
>
> The layer-specific $c_t$ allows specific layers to have small regularization and other layers to have large regularization. This gives SPD the edge over other methods by fitting the new dataset (better ID performance) and retraining knowledge from the pre-trained model (better OOD performance). We will update the description to make this point more clear.
>
> **The statement "We recommend starting with and adjusting the strength according to the specific needs" is rather unsatisfying and doesn't provide the reader with a good idea of whether or not is easy to tune. This can probably be resolved by adding additional analysis along this end compare do just Table 2.**
>
> Thank you for mentioning the ease of tuning this method. This is a contribution of this work. Practically, the method is just as easy to tune as regular weight decay and L2 regularization because the hyper-parameter $\lambda$ in SPD has the same effect as in those methods, i.e., adjusting regularization strength. In some cases, it can be much easier to tune as shown in the sensitivity analysis in Table 2 of main paper. SPD is much less sensitive to the hyperparameter, consistently yielding good performance on both ID and OOD datasets, whereas L2-SP is more sensitive to the choice of hyperparameter.
>
> Moreover, SPD is even simpler to understand as described in Section 3.4. We re-parametrized L2 regularization (weight decay) as a projection. This gives an intuitive interpretation of the regularization hyper-parameter. For example, in a conventional weight decay setup, we can set the hyper-parameter as $\lambda=0.01$. We do not necessarily know what $0.01$ means as a hyper-parameter. However, with the re-parametrization in Section 3.4, we can set it to be $\lambda=1$; this intuitively means that the regularization projects the current model weights back to the deviation of the last update.
>
>
> **Experimentally, the results feel somewhat lacking and somewhat misleading. For example, in the Domain-Net experiments, the only comparison is with AdamW, which inherently should lack the exploration capabilities the authors are attempting to account for. There are quite a few baselines that are missing as a result, in my opinion, such as sharpness aware minimization (SAM) and/or other exploration-based optimizers.**
>
> Thank you for bringing up this work. We have added a comparison to SAM on ImageNet (Table 2 in the rebuttal PDF). SPD outperforms SAM in all experiments. SPD is designed explicitly for robust fine-tuning of pre-trained foundation models, whereas SAM does not consider pre-trained initialization. SAM focuses on finding a good local minimum with uniformly low loss. This property could lead to better generalization for regular training but not necessarily a more robust fine-tuned model. Fine-tuning requires careful balancing between fitting the new data and retaining knowledge in the pre-trained model. SPD explicitly considers this problem.
>
> There may be other exploration-based optimizers to achieve better generalization. However, they are not designed for fine-tuning and generally do not consider the pre-trained initialization in their formulation. Instead, our experiments focus on comparing baselines for robust fine-tuning (Tables 3 and 4 in the main paper). In our setup, AdamW is the best example of an exploration-enhanced optimizer. It is equipped with an adaptive learning rate and momentum. Both mechanisms encourage AdamW to escape local minima and explore further. However, our understanding of exploration may differ from that of the reviewer. We welcome further discussion.
>
>
> **Although not a deal-breaker, I would appreciate a perhaps more in-depth analysis of the language modelling experiments, in particular without PEFT. While I understand the relationship with PEFT and the proposed optimizer, I believe that it would be quite valuable to show how AdamW-SPD works even under naive fine-tuning scenarios.**
>
> We provide a new experiment for VQA tasks. Specifically, we fine-tune the PaliGemma-3B model on the VQAv2 dataset with LoRA and test it with nine additional VQA datasets consisting of different types of distribution shifts across vision, question, and answer (IV-VQA, CV-VQA, VQA-Rephrasings, VQA-CE, AdVQA, TextVQA, VizWiz and OK-VQA). In Tab.~\ref{tab:vqa} of the rebuttal PDF, SPD achieves the best ID and average OOD performance. We also show the performance evaluation for both near and far OOD datasets. SPD is consistently more robust under different types and degrees of distribution shifts.
>
> We acknowledge that full fine-tuning of large language models using SPD would be very interesting. However, our experiments are limited by computation resources and time. It will be an interesting next step in our exploration.

---

> > ### Comment · Reviewer_CVio · 2024-08-08
> >
> > I appreciate the response provided by the authors. I believe the authors have addressed most of my concerns and I have adjusted my score accordingly. While I am now more convinced about the merits of the work, I still believe that increased analysis of the proposed method (from a more theoretical perspective rather than completely empirical) would be more meaningful given that the nature of the work.

---

### Official Review · Reviewer_Ex5N · 2024-07-13

**Soundness:** 3
**Presentation:** 4
**Contribution:** 3
**Rating:** 7
**Confidence:** 4

**Summary:**

This paper proposes a novel weight decay strategy called Selective Projection Decay (SPD).
SPD selectively imposes a stronger penalty on certain layers, and is designed to improve both in-distribution (ID) and out-of-distribution (OOD) performance.
The paper demonstrates the effectiveness of SPD through experiments on several benchmarks.

**Strengths:**

- SPD is well-motivated and simple to implement.
- The paper is well-written and easy to follow. It clearly lays out intuition and motivation. I especially liked sections 3.3 and 3.4, which relate the condition to online hyperparameter optimization and the deviation ratio to a re-interpretation of L2-SP as a projection. I thought the line of logic here was quite clear.
- The paper demonstrates strong performance on several standard benchmarks for robust fine-tuning.

**Weaknesses:**

- Overall, I think this is a strong paper, and its strengths outweigh its weaknesses.
I think the paper could benefit from an ablation experiment, for example, trying Adam-SPD without the condition or benchmarking SPD's performance with optimizers other than Adam.
- [1] reports that ensembling with the initial weights (whether in model space or weight space) is a simple strategy that improves OOD robustness, and a few recent works [2, 3] report that ensembling continues to improve performance after their proposed fine-tuning strategies. Could the authors validate how the benefits from SPD extend to the ensembling setting?

[1] Robust fine-tuning of zero-shot models
[2] Finetune like you pretrain: Improved finetuning of zero-shot vision models
[3] AutoFT: Learning an Objective for Robust Fine-Tuning

**Questions:**

Please see the weaknesses section.

**Limitations:**

Please see the weaknesses section.

---

> ### Author Response · Authors · 2024-08-06
>
> We really appreciate the positive comments and suggestions for ensemble methods. In the response, we clarified your concerns and included new experiments. If anything is not clear, we are happy to discuss it during the rebuttal period.
>
> **Overall, I think this is a strong paper, and its strengths outweigh its weaknesses. I think the paper could benefit from an ablation experiment, for example, trying Adam-SPD without the condition or benchmarking SPD's performance with optimizers other than Adam.**
>
> Thank you for the positive comments on this work. The ablation study without the condition is shown in Table 2. Essentially, without the selecting condition (line 96-102), SPD reduces to a regular L2 regularization, which is shown ineffective in our comprehensive analysis (line 255). Furthermore, SPD is a general method and applicable to other optimizers. We hypothesize that it should have positive effects on optimizers using regular weight decay and L2 regularization.
>
>
> **[1] reports that ensembling with the initial weights (whether in model space or weight space) is a simple strategy that improves OOD robustness, and a few recent works [2, 3] report that ensembling continues to improve performance after their proposed fine-tuning strategies. Could the authors validate how the benefits from SPD extend to the ensembling setting?**
>
> Thank you for mentioning these works, especially the ensemble technique such as Wise-FT. Technically, ensemble techniques are orthogonal methods. SPD should further improve the performance of them such as WISE-FT. To validate this, we added experiments combining WISE-FT and SPD in our Imagenet experiments (Table 1 in the rebuttal PDF). Our results show that SPD further improves the performance of Wise-FT. It's also worth pointing out that SPD is more general than WISE-FT, which only applies to models with linear connectivity and capable of zero-shot classification. It does not apply to the segmentation experiment (Table 4 in the main paper), whereas SPD does not have this constraint.

---

### Official Review · Reviewer_dDi9 · 2024-07-15

**Soundness:** 2
**Presentation:** 2
**Contribution:** 2
**Rating:** 6
**Confidence:** 4

**Summary:**

This work proposes a new regularization scheme called Selective Projection Decay (SPD), which selectively imposes penalties on layers where the current progress direction does not align with the vanilla update direction. This method is compatible with PEFT methods, including LoRA-type algorithms, making it practical for end users. Experiments show that ADAM equipped with SPD consistently outperforms baselines, including L2 regularization, in terms of in-distribution generalization and out-of-distribution robustness on multiple popular vision and language benchmarks.

**Strengths:**

The simplicity of the proposed method is its major strength, making it easy for many users to adopt. Compatibility with parameter-efficient fine-tuning is a significant advantage. Additionally, its simplicity ensures good reproducibility. Moreover, the experiment details are well described, further supporting reproducibility.

The presentation is very clear. Mostly, it’s easy to follow the main idea, interpretations, and experiment results. Specifically, the difference between L2-SP and SPD is clearly explained by comparing the pseudo-codes and providing easy-to-understand interpretations.

**Weaknesses:**

No obvious weaknesses are observed. One minor issue is that the performance improvement, while consistently observed, is not significantly high. Additionally, I have noted some unclear justifications in the manuscript, as described in the questions section.

**Questions:**

#1. How does the performance compare when using EWC regularization for fine-tuning, since EWC is generally better than L2 regularization?

#2. Comparing the performance of L2-SP and Adam-SPD in hyper-parameter sweeping experiments seems challenging since these are reported in tables. Perhaps heatmaps could be a clearer alternative to visualize and highlight the results.

#3. The authors argue that SPD selectively imposes penalties on large deviations in a layer-wise manner. However, in Algorithm 2, there is no layer-wise information. It appears that all parameters are updated based on the alignment score denoted by c_t. From my understanding of Algorithm 2, if the alignment is high (i.e., c_t is smaller than zero), the updated parameters are further adjusted by the interpolation-like equation. If c_t is larger than zero, do we use the vanilla ADAM update, since \theta_t has already been updated before calculating c_t? Am I correctly understanding Algorithm 2? In such cases, it seems that the parameters shouldn’t be updated, since SPD will impose a penalty to slow down updates for those layers with negative c_t.

#4. In the LLaMA PEFT experiments, there is no baseline with L2 regularization or its advanced versions (such as EWC). While I expect that ADAM with SPD is better than AdamW, I would like to see if there are also improvements compared to ADAM with L2 regularization in PEFT settings.

**Limitations:**

The authors state that the proposed approach only works well if the pretrained model is good enough to provide good initial parameters. No clear negative societal impacts have been observed.

---

> ### Author Response · Authors · 2024-08-06
>
> We thank the reviewer for the positive comments and for suggesting insightful new related works. In the response, we clarified all your concerns. If anything is not clear, we are happy to discuss it during the rebuttal period.
>
> **How does the performance compare when using EWC regularization for fine-tuning, since EWC is generally better than L2 regularization?**
>
> Thank you for mentioning this work!  However, EWC does not apply to general fine-tuning and was explicitly developed for continual learning (CL). Under the CL setting, EWC requires the gradient information from the previous task, which is not available for fine-tuning a pre-trained model. Therefore, we cannot fairly compare to EWC in the fine-tuning setting. Nevertheless, EWC  points out that uniform regularization can be ineffective, which is also the main motivation of this paper for fine-tuning.
>
>
> **Comparing the performance of L2-SP and Adam-SPD in hyper-parameter sweeping experiments seems challenging since these are reported in tables. Perhaps heatmaps could be a clearer alternative to visualize and highlight the results.**
>
> Yes, this is a very good suggestion. We will update the tables into heat maps in the main paper.
>
> **The authors argue that SPD selectively imposes penalties on large deviations in a layer-wise manner. However, in Algorithm 2, there is no layer-wise information. It appears that all parameters are updated based on the alignment score denoted by $c_t$. From my understanding of Algorithm 2, if the alignment is high (i.e., $c_t$ is smaller than zero), the updated parameters are further adjusted by the interpolation-like equation. If $c_t$ is larger than zero, do we use the vanilla ADAM update, since $\theta_t$ has already been updated before calculating $c_t$? Am I correctly understanding Algorithm 2? In such cases, it seems that the parameters shouldn’t be updated, since SPD will impose a penalty to slow down updates for those layers with negative $c_t$.**
>
> SPD uses layer-wise information in the calculation of $c_t$. Specifically, each layer has a different $c_t$. Here, $\theta_t$ denotes the weight matrix of a layer, not an individual parameter or the entire network.  We will make the notation clear in the update. Yes, when $c_t>0$, the layer $\theta_t$ is updated using vanilla Adam. If this does not clarify your confusion, we are happy to discuss this more in the following rebuttal period.
>
>
> **In the LLaMA PEFT experiments, there is no baseline with L2 regularization or its advanced versions (such as EWC). While I expect that ADAM with SPD is better than AdamW, I would like to see if there are also improvements compared to ADAM with L2 regularization in PEFT settings.**
>
> In the LLaMA PEFT experiments, we compared our method with Adam $+$ weight decay, as indicated in the table title. In this PEFT setting, L2 regularization reduces to weight decay as explained in section 3.5. In other words, the L2 regularization baseline is already included. We will make this clearer in the text; thanks for bringing this up.

---

### Author Rebuttal · Authors · 2024-08-06

We thank all of the reviewers for their positive comments on this work's adaptability (dDi9), performance (Ex5N, CVio, Jrpf, 33UR), and insights (Ex5N, 33UR). We aim to provide concrete responses to your questions and clarify your confusion.

The rebuttal PDF includes new experiments and studies requested by the reviewers. The new experiments are summarized below.

*  (Jrpf,33UR) We added L2-SP and Infograph to Table 1 in the main paper. Our method remains the best-performing method.
* (CVio) We added a comparison to Sharpness-Aware-Minimization (SAM) on ImageNet . SPD outperforms SAM on ImageNet because SAM is not explicitly designed for robust fine-tuning.
* (CVio) We added a new VQA fine-tuning experiment with 9 additional VAQ OOD datasets with distribution shifts across vision and language. SPD is consistently more robust under different types and degrees
of distribution shifts.

We hope our response answers your questions and are open to further discussion during the rebuttal period.

---

### Decision · Program_Chairs · 2024-09-25

**Decision:**

Accept (poster)

**Comment:**

The paper introduces a regularisation technique for finetuning large pretrained models by imposing penalties on specific layers to enhance both ID and OOD generalisation.

The reviewer consensus is an acceptance. They appreciate the simplicity of the method, clear presentation and strong empirical performance. There were some remaining reservations:
- CVio: I still believe that increased analysis of the proposed method (from a more theoretical perspective rather than completely empirical) would be more meaningful given that the nature of the work.
- Jrpf: I agree with the other reviewer that the theoretical perspective is lacking in this work, though the empirical results are strong.

The merits greatly outweigh the concerns in this case. We recommend an acceptance.